# Human fetal cartilage-derived chondrocytes and chondroprogenitors display a greater commitment to chondrogenesis than adult cartilage resident cells

Elizabeth Vinod[1,2]*, Ganesh Parasuraman[2], Jeya Lisha J.[1], Soosai Manickam Amirtham[1], Abel Livingston[3], Jithu James Varghese[4], Sandya Rani[2], Deepak Vinod Francis[5], Grace Rebekah[6], Alfred Job Daniel[3], Boopalan Ramasamy[7,8]*, Solomon Sathishkumar[1]*

1 Department of Physiology, Christian Medical College, Vellore, India, 2 Centre for Stem Cell Research, (A Unit of InStem, Bengaluru), Christian Medical College, Vellore, India, 3 Department of Orthopaedics, Christian Medical College, Vellore, India, 4 Department of Diabetes, School of Life Course Sciences, King's College London, London, United Kingdom, 5 Department of Anatomy, Christian Medical College, Vellore, India, 6 Department of Biostatistics, Christian Medical College, Vellore, India, 7 Faculty of Health and Medical Sciences, The University of Adelaide, Adelaide, Australia, 8 Department of Orthopaedics and Trauma, Royal Adelaide Hospital, Adelaide, Australia

* elsyclarence@cmcvellore.ac.in (EV); boopalan.ramasamy@sa.gov.au (BR); solomon@cmcvellore.ac.in (SS)

## Abstract

Obtaining regeneration-competent cells and generating high-quality neocartilage are still challenges in articular cartilage tissue engineering. Although chondroprogenitor cells are a resident subpopulation of native cartilage and possess a high capacity for proliferation and cartilage formation, their potential for regenerative medicine has not been adequately explored. Fetal cartilage, another potential source with greater cellularity and a higher cell-matrix ratio than adult tissue, has been evaluated for sourcing cells to treat articular disorders. This study aimed to compare cartilage resident cells, namely chondrocytes, fibronectin adhesion assay-derived chondroprogenitors (FAA-CPCs) and migratory chondroprogenitors (MCPs) isolated from fetal and adult cartilage, to evaluate differences in their biological properties and their potential for cartilage repair. Following informed consent, three human fetal and three adult osteoarthritic knee joints were used to harvest the cartilage samples, from which the three cell types a) chondrocytes, b) FAA-CPCs, and MCPs were isolated. Assessment parameters consisted of flow cytometry analysis for percentage expression of cell surface markers, population doubling time and cell cycle analyses, qRT-PCR for markers of chondrogenesis and hypertrophy, trilineage differentiation potential and biochemical analysis of differentiated chondrogenic pellets for total GAG/DNA content. Compared to their adult counterparts, fetal cartilage-derived cells displayed significantly lower CD106 and higher levels of CD146 expression, indicative of their superior chondrogenic capacity. Moreover, all fetal groups demonstrated significantly higher levels of GAG/DNA ratio with enhanced uptake of collagen type 2 and GAG stains on histology. It was also noted that fetal FAA CPCs had a greater proliferative ability with significantly higher levels of the

**Data Availability Statement:** All relevant data are within the paper and its Supporting Information files.

**Funding:** This project was supported by the 1. Department of Biotechnology (BT/PR32777/MED/31/415/2019), Govt. of India Recipient: E.V 2. Fluid Research Grant (IRB Min No: 14498 dated 23.02.2022), Christian Medical College, Vellore. Recipient: S.M.A There was no additional external funding received for this study. The funders had no role in study design, data collection and analysis, decision to publish or preparation of the manuscript.

**Competing interests:** The authors have declared that no competing interests exist.

primary transcription factor SOX-9. Fetal chondrocytes and chondroprogenitors displayed a superior propensity for chondrogenesis when compared to their adult counterparts. To understand their therapeutic potential and provide an important solution to long-standing challenges in cartilage tissue engineering, focused research into its regenerative properties using in-vivo models is warranted.

## Introduction

Articular cartilage displays low self-repair and regenerative capacity due to its relative avascularity, low cellularity, and a small number of progenitor cells [1]. Cell-based therapy has been utilised as a mainstay treatment option for cartilage afflictions such as osteoarthritis and chondral defects. The foremost contenders, used either as stand-alone substitutes or self-contained with bio scaffolds, are chondrocytes and Mesenchymal Stem Cells (MSCs) [2]. Although autologous chondrocyte transplantation shows reasonable outcomes as a cartilage repair strategy, its major shortcoming is the minimal cartilage available for harvest, which requires a surgical procedure. The low number of released chondrocytes from the harvested cartilage mandates extended monolayer expansion in culture, leading to dedifferentiation. Studies using various MSCs display advantages due to ease of availability; however, their increased inclination for osteogenesis and the formation of fibrocartilage in-vivo have limited their use for chondrogenesis [3,4].

The search for an alternative and optimal cell type led to the discovery of a potential cell residing within the articular cartilage among chondrocytes called chondroprogenitors (CPCs) [5]. Isolated by migratory and fibronectin adhesion assay, articular cartilage-derived CPCs have been categorised as MSCs demonstrating distinct surface marker expression and multilineage potential [6]. The standard method for the isolation of CPCs includes the subjection of chondrocytes to fibronectin adhesion assay, which enables enrichment for colony forming cells based on their expression of the fibronectin integrin receptor followed by clonal expansion [7,8]. Another established method for isolating CPCs includes culturing and strategically using cartilage explants based on their migratory potential [9,10]. In-vitro studies using fibronectin adhesion assay-derived chondroprogenitors (FAA-CPCs) have reported their superiority as a potential therapeutic, as they display a higher expression of chondrogenic genes, such as SOX-9 and lubricin, and lower levels of hypertrophy markers, such as RUNX2 and Collagen type X, as compared to chondrocytes and Bone Marrow (BM) MSCs [11–13]. Similarly, migratory chondroprogenitors (MCPs) display higher migratory ability and chondrogenic potential when compared to MSCs and chondrocytes [9,14,15]. A recent in-vitro report comparing FAA-CPCs to MCPs shows that the latter progenitor population retains and displays superior cartilage repair under normoxia culture conditions [16]. The promising in-vitro results displaying predilection for hyaline-like regeneration led to some preclinical in-vivo studies. Both FAA-CPCs and MCPs report the ability to attenuate OA and heal osteochondral defects, demonstrating superior repair tissue as compared to BM-MSCs [17–22].

The cells isolated from fetal cartilage, another potential source with higher cellularity containing a higher cell-matrix ratio than adult tissue, have been evaluated for treating articular disorders [23–25]. Few studies have attempted to use fetal cartilage-derived cells to assess their potential for cartilage repair. Fuchs et al. report that ovine fetal chondrocytes derived from hyaline or elastic cartilage have a higher proliferative capacity, expressing significantly higher glycosaminoglycans (GAG) levels and collagen type II than adult chondrocytes [25]. In

another comparative study using fetal and adult BMMSCs, the proliferative and differentiation potential of fetal BMMSCs was reported to be higher [26]. Choi et al. report that as compared to BM-MSCs and young adult donor chondrocytes, 12-week gestation human fetal cartilage chondrocytes showed more chondrogenic ability even at extended passages, with low senescence and maintained SOX-9 expression [24]. Additionally, in-vitro three-dimensional pellet culture and in-vivo studies using polyglycolic acid scaffolds showed that fetal chondrocytes exhibited more significant chondrogenic potential than the BM-MSCs and young chondrocytes. Further, a clinical trial assessing the application of allogeneic fetal chondrocytes as a treatment in rheumatic arthritis showed the desired clinical effect by significantly reducing the C-reactive protein, erythrocyte sedimentation rate, gamma globulin and fibrinogen over one year with no further complications [27]. Thus, fetal tissue has shown regenerative healing capabilities in an adult environment without eliciting an allogeneic response or rejection [28,29]. No studies have so far evaluated fetal chondroprogenitors derived by fibronectin adhesion assay or migratory assay and compared it to adult chondrocytes and adult chondroprogenitors.

Our study aimed to compare the cartilage resident cells, namely chondrocytes, FAA-CPCs and MCPs isolated from fetal or adult cartilage, to evaluate differences in their biological attributes and prospects for cartilage repair. In addition, we aimed to examine the fetal cartilage subpopulations to infer the suitable cell source for cartilage repair.

## Methods and materials

### Sample procurement and study design

The Institutional Review Board (Christian Medical College, Vellore, India, IRB Min No: 14498 dated 23.02.2022) approved the research protocols for this study. All methods used ensured compliance with all relevant guidelines and regulations. For the adult human cartilage samples, written informed consent was obtained from three patients who required total knee replacement as part of their treatment for high-grade osteoarthritis [n = 3, age: 63±14 years (mean ± SD)]. The joints were harvested from patients having a Kellgren-Lawrence radiological score of 4, whereas indications for exclusion included tumours, infection or inflammation. The articular cartilage shavings were harvested from non-weight bearing areas containing preserved full depth cartilage. Fetal cartilage samples were obtained from the knee of fetuses (n = 3, female fetus, GA: 36+4, 27+3, 23+3 weeks) that had been spontaneously aborted or terminated due to medical reasons after obtaining written informed consent from the parents.

Once harvested, the adult and fetal cartilage slices were subjected to cellular enzymatic digestion to release the chondrocytes. To obtain the FAA-CPCs subset, the chondrocytes were loaded onto fibronectin-coated plasticware, and the obtained polyclones were further expanded. To isolate the MCPs, standard cartilage explants were cultured in an expansion medium to enable outgrowth of the migratory subset, which was further grown to sub confluence (80% confluence). This study involved six groups for comparison, where adult and fetal joint-derived chondrocytes, FAA-CPCs and MCPs at passages 1–2 were subject to phenotypic characterisation to assess their mesenchymal, chondrogenic and endochondral ossification properties. Several parameters were evaluated, including fluorescence-activated single cell sorting (FACS) for percentage expression of surface cell markers, population doubling time and cell cycle analyses for growth kinetics, qRT-PCR for markers of chondrogenesis and hypertrophy, trilineage differentiation and biochemical analysis of differentiated chondrogenic pellets for total GAG/DNA content, (Fig 1). The conducted experiments included three biological samples (n = 3).

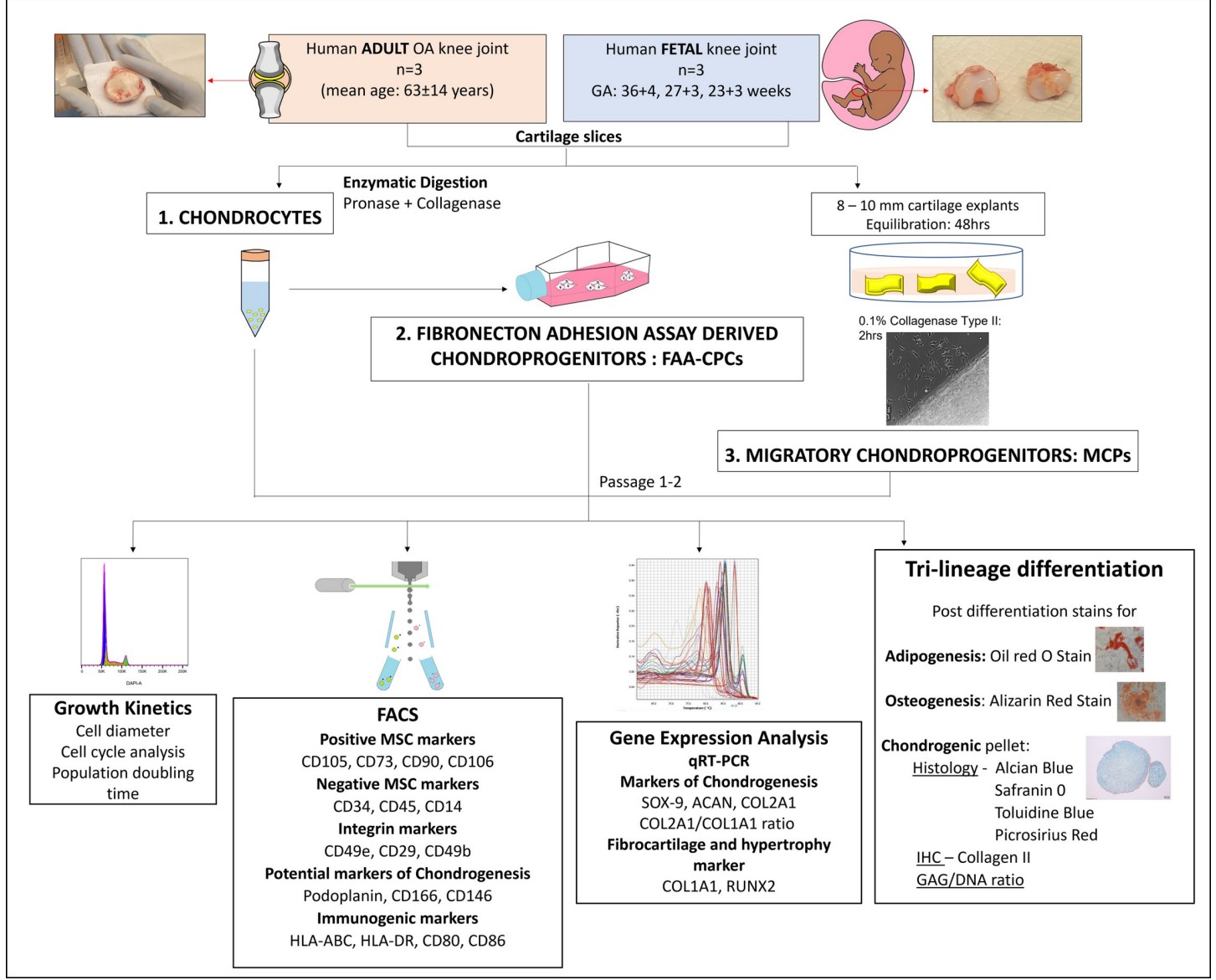

**Fig 1. Study algorithm depicting the six groups used for comparison and their evaluation parameters.** The six groups included were adult and fetal cartilage derived chondrocytes, fibronectin adhesion assay-derived chondroprogenitors (FAA-CPCs) and migratory chondroprogenitors (MCPs). OA: Osteoarthritis, CD: Cluster of differentiation, SOX-9: (Sex-determining region Y)-box 9, ACAN: Aggrecan, COL: Collagen, RUNX2: Runt-related transcription factor-2, MMP-13: Matrix metalloproteinase-13, GAG: Glycosaminoglycans. All experiments were performed with three biological samples (n = 3). The included figure is similar but not identical to the original image and is for illustrative purposes only.

## Isolation and culture of cell

**Adult and fetal chondrocytes.** Cartilage shavings were washed with phosphate-buffered saline to remove traces of synovial fluid, were minced to a size of 2-3mm$^2$ and subjected to sequential overnight cellular digestion using the enzymes pronase (12 IU, Roche) for 180min followed by collagenase type II (100 IU, Worthington) for 12hr in a shaking water bath. Five thousand cells/cm$^2$ were loaded and expanded with DMEM/F12 containing 10% fetal calf serum (FCS, Thermo Fisher Scientific) and antibiotics.

**FAA-CPCs.** To isolate FAA-CPCs, the enzymatically released chondrocytes at a loading concentration of 4000 cells/9.3cm2 were added to fibronectin-coated plates (10μg/ml, Sigma) for 20 min. The removal of non-adherent chondrocytes followed this. The adherent cells were continued in culture for another 12 days till each attached cell achieved five population doublings (each clone > 32 cells). The enriched polyclonal FAA-CPCs were isolated, replated, and expanded in a stromal medium containing 1 ng/ml of recombinant transforming growth factor beta 2 and 5ng/ml human fibroblastic growth factor-2 at 5ng/ml, as per standard established protocol [7].

**Migratory chondroprogenitors (MCPs).** 8–10 mm of articular cartilage from the same joint was harvested and placed in an expansion medium containing 10mM Glutamine, antibiotics, and antimycotics. Isolation protocol was followed as described by Koelling et al [9]. In brief, the cartilage explants following an equilibration for 48 hr were placed in collagenase type II solution (0.1%, Worthington) at 37˚C for a period of 2 hr. Further, the cartilage slices were washed to remove any released cells, were placed in a culture plate containing expansion medium and observed for the outgrowth of cells. The migrated cells following confluency were harvested, further expanded for phenotypic characterisation.

The medium change was conducted every third day for all cultures and harvested using 0.125% of Trypsin-EDTA (GIBCO, Thermo Fisher Scientific) at 85–90% confluence. Expansion and differentiation of the six groups (adult and fetal chondrocytes, FAA-CPCs and MCPs) were performed under their specified culture conditions. Since the standardised culture conditions recommend the requirement of additional growth factors for FAA-CPCs but not for chondrocytes and MCPs, the three types of cells were cultured accordingly.

## Population doubling time, cell diameter and cell cycle analysis

All six study groups were grown in monolayer culture with a loading density of 5000 cells/cm2 and passaged till 80% confluence. Cell count was performed using a cell counter (Cell Drop BF, De Novix) by the trypan blue exclusion technique. The population doubling time (PDT) for all the groups was calculated using the following formulae:

$$PDT = \left(\log_2 x \text{ days in culture}\right) / \log(FD) - \log(ID)$$

Where FD: the number of cells obtained at 80% confluence and ID: the initial number of loaded cells. PDT between the groups was compared from passage 1 to passage 2.

DAPI (4',6-diamidino-2-phenylindole) was used to define the percentage of cells in distinct cell cycle phases. The cells were trypsinised at 60–70% confluence, and an automated cell counter estimated the cell size at passage1-2 (Countess, Invitrogen). The harvested cells were fixed with 70% ice-cold ethanol for 60 min. After a wash, 1μg/ml of DAPI was added for 30 minutes were the cells washed and the cells resuspended for flow cytometry analysis using BD FACS Celesta flow cytometer using BD FACSDiva Software Version 8.0.1.1. Cell cycle analysis was done using Flow-Jo v10.8.1 software with Dean Jett algorithm

## Fluorescence Activated Cell Sorting (FACS): Surface marker expression

At sub confluence (80%), cells from the six groups (biological replicates, n = 3) were harvested, washed and processed for flow cytometry. Staining was conducted as described in the technical data sheet provided by the manufacturer with each antibody. Since the progenitors report a similar profile as MSCs, the first group of surface markers assessed for comparison consisted of CD105-FITC, CD73-PE, CD90-PE, CD106-APC (markers of positive expression), and CD34-PE, CD45-FITC, and CD14-FITC (markers of negative expression). Secondly, the integrin markers CD 49e-PE, CD29-APC, and CD49b-FITC were compared. In the third group,

CD markers reported to be potential markers of enhanced chondrogenesis were assessed: CD166-BB515, CD146-PE, and Podoplanin-BV421. HLA-ABC class I(PE) (HLA-ABC-PE), HLA-DR class II (V500), and their co-stimulatory molecules, CD80 BB515 and CD86 BV421, were included in the final group (Tables 1 and S1). BC CytoFLEX LX flow cytometer and CYTExpert Software Version 2.5 was used for acquisitions, and analysis was done using FlowJo v10.8.1 software

## Quantitative Reverse Transcriptase-Polymerase Chain Reaction (qRT-PCR)

In accordance with the manufacturer's instructions, the harvested cells were washed in PBS, and RNA was extracted with the Qiagen RNeasy Mini Kit. The nanodrop spectrophotometer was used to determine the RNA concentrations at 260 and 280 nanometers (A260/A280). By using the Takara Bio First-Strand synthesis system, complementary DNA was synthesised with 280 ng of RNA. A QuantStudio 6K Flex thermocycler (Applied Biosystems) was used to conduct PCR assays with Low Rox Takyon SYBR Master Mix Dttp (Eurogentec, Belgium). The gene profile included assessing expression of markers for chondrogenesis (*SOX-9*, *ACAN*, and *COL2A1*), fibrocartilage (*COL1A1*), and hypertrophic chondrocytes (*RUNX2*). The relative mRNA expression was normalised for each target gene to *GAPDH*, the housekeeping gene. The individual ΔCt values were compared to the fetal chondrocyte group (ΔΔCt), from which 2^-ΔΔCt was calculated. Additionally, the functional ratio of *COL2A1* to *COL1A1* was calculated to assess their ability to form hyaline cartilage. A total of three biological samples were examined in two technical replicates (n = 3). Detailed primer sequences, accession codes, gene identifiers, and base pair length are listed in S2 Table.

## Multilineage differentiation and confirmatory staining

StemPro differentiation kits (Thermo Fisher, Cat no. A1007201, A1007001, and A1007101) were used to induce chondrogenic, osteogenic, and adipogenic differentiation. To initiate adipogenic and osteogenic differentiation, cells were seeded in 24 well culture dishes at 5000 cells/cm2, expanded to sub confluence, and replaced with the differentiation media. The control arm included cells grown in expansion medium for the same duration. For 3D pellet cultures, $1 \times 10^6$ cells were loaded into 15ml centrifuge tubes, spun at 400g for 12 minutes, and left uninterrupted for 48 hr. The pellets were supplemented with chondrogenic medium every three days for a three-week period.

**Adipogenic and osteogenic staining.** Following adipogenic differentiation, the cells were fixed with 10% formaldehyde, washed, and stained with 0.5% Oil Red O (Sigma). In comparison the differentiated osteogenic cells were fixed with 70% ethanol, washed and stained with 2% Alizarin Red (Sigma). Parallel controls cultured on a stromal medium were also stained, and all images were captured using the Olympus virtual slide system.

**Chondrogenic confirmatory staining.** After 21 days of differentiation, the pellets were subjected to fixation using 4% paraformaldehyde, paraffin embedded, sectioned (4 μm) and subjected to confirmatory staining. The protocol applied for the individual stains was as follows: for assessing glycosaminoglycan accumulation, the sections were incubated with Alcian blue (pH:2.5, Cat no: J60122, Alfa Aesar, US), for 5 min and counterstained with neutral red, whereas for Safranin O fast green staining, Wiegert's Iron Hematoxylin followed by subsequent incubation with acid alcohol (1%), fast green solution (0.05%), acetic acid (1%) and Safranin O solution (1%) was used. Toluidine blue staining was performed by incubating 0.1% of the dye solution for 5 min (C.I. 52040 Qualigens). PicroSirius red (C.I.35782) staining was performed using a 0.1% dye concentration, and then a counterstain of Hematoxylin was applied.

**Table 1. Fluorescence-activated cell sorting data.** The groups included: Positive and negative MSC markers, potential markers of enhanced chondrogenesis, and immunogenic markers of the six cell groups. Data is expressed as percentage mean ± Standard Deviation (n = 3).

| Groups | | Fetal Chondrocytes | Fetal FAA-CP | Fetal MCP | Adult chondrocytes | Adult FAA-CP | Adult MCP |
|---|---|---|---|---|---|---|---|
| **Positive MSC markers** | **CD105** | 89.88±10.57 | 86.32±5.54 | 81.51±20.14 | 96.35±2.22 | 99.1±0.82 | 97.73±3.20 |
| | **CD73** | 99.97±0.03 | 99.93±0.10 | 99.97±0.02 | 99.97±0.03 | 99.98±0.01 | 99.96±0.02 |
| | **CD90** | 99.99±0.01 | 99.99±0.01 | 99.98±0.02 | 99.91±0.07 | 98.80±2.02 | 99.93±0.10 |
| | **CD106** | 8.68±5.69 | 6.26±3.65 | 28.69±8.75 | 58.38±4.76 | 53.02±17.94 | 54.41±16.78 |
| **Negative MSC markers** | **CD34** | 14.24±7.45 | 6.90±6.37 | 4.96±3.54 | 6.45±4.76 | 5.93±5.08 | 1.6±0.36 |
| | **CD45** | 0.26±0.37 | 0.11±0.07 | 1.14±1.23 | 0.39±0.67 | 0.48±0.34 | 3.13±2.06 |
| | **CD14** | 0.15±0.21 | 4.64±7.75 | 0.75±0.74 | 0.06±0.05 | 0.29±0.11 | 0.94±0.81 |
| **Integrin markers** | **CD49e** | 99.98±0.03 | 99.98±0.02 | 99.95±0.05 | 99.97±0.02 | 99.99±0.01 | 99.96±0.02 |
| | **CD29** | 99.95±0.01 | 99.95±0.05 | 99.96±0.04 | 99.93±0.01 | 99.95±0.03 | 99.92±0.02 |
| | **CD49b** | 7.18±10.55 | 24.56±18.45 | 9.12±12.03 | 4.77±2.78 | 37.47±4.16 | 15.44±2.76 |
| **Potential markers of chondrogenesis** | **Podoplanin** | 97.36±1.05 | 94.32±6.69 | 96.29±4.43 | 91.55±7.60 | 91.24±7.05 | 89.64±11.51 |
| | **CD166** | 99.36±0.67 | 99.92±0.03 | 99.5±0.23 | 98.92±0.21 | 99.97±0.04 | 99.86±0.01 |
| | **CD146** | 99.5±0.24 | 99.91±0.02 | 99.28±0.92 | 41.42±15.30 | 70.71±13.50 | 61.09±14.97 |
| **Immunogenic markers** | **HLA-ABC** | 99.89±0.15 | 99.97±0.03 | 99.78±0.30 | 96.83±5.32 | 99.97±0.03 | 99.99±0.01 |
| | **HLA-DR** | 0.20±0.32 | 0.13±0.20 | 0.55±0.59 | 2.50±3.31 | 0.26±0.24 | 0.99±0.69 |
| | **CD80** | 0.74±0.84 | 0.11±0.10 | 1.16±1.56 | 1.68±0.42 | 0.93±1.24 | 4.37±1.60 |
| | **CD86** | 0.46±0.71 | 0.17±0.27 | 0.63±0.63 | 2.89±3.59 | 0.25±0.36 | 1.40±0.97 |

Following staining, all slides were dehydrated, cleared using xylene and mounted with DPX for imaging.

**Immunohistochemistry: Collagen type II (Chondrogenic differentiated pellet).** The chondrogenic differentiated paraffin pellet sections were subjected to antigen retrieval using pronase (1mg/mL) and hyaluronidase (2.5mg/mL). At 4°C, the sections were subject incubated with mouse monoclonal anti-collagen type II antibodies (5μg/mL) (DSHB, II-II6B3), followed by a 30-minute incubation with a 1:250 secondary antibody: HRP labelled goat anti-mouse immunoglobulin (31430, Pierce, Wisconsin, USA), then stained with a 3,3-diamino-benzidine solution followed by counterstaining with Hematoxylin. Histological imaging of the stained sections was carried out using an Olympus BX43f microscope.

**Biochemical analysis of chondrogenic pellets: Total GAG/DNA content.** Chondrogenically differentiated pellets were subjected to papain digestion (120 g/ml at 65°C, 16 h) to determine the GAG and DNA content. Lambda DNA was used to generate a standard curve, and Quant-iT Picogreen dsDNA reagent was utilised to measure the DNA concentration. The fluorescence intensity (Ex λ: 480nm, Em λ: 520 nm) was measured with a SpectraMax i3x Reader (Norwalk, CT, USA). Dimethyl methylene blue dye was used to determine GAG content using chondroitin 6-sulphate as a standard (Chondrex, Inc Cat No: 6022). An Elisa plate reader was used to determine the optical density (Absorbance: 525nm). The GAG content was then normalised to DNA content to obtain the GAG/DNA ratio.

## Statistical analysis

Data were analysed using SPSS 21.0. IBM Bangalore, and Prism v6 (GraphPad) was used for graphical representation. All values were depicted as mean ± standard error mean. In order to compare the six groups, a one-way ANOVA was applied after determining the normality of the distribution. LSD post hoc test was conducted for intergroup comparisons and statistical significance was determined by a P-value of 0.05.

# Results

## Growth kinetics: Cell diameter and cell cycle analysis

Fetal chondrocytes, Fetal and adult FAA-CPCs demonstrated clonal growth, with the FAA-CPCs achieving a population doubling of 5, namely, each clone containing more than 32 cells by day 10–11 (Fig 2A, 2C and 2I). MCPs migrating from cartilage explants were observed as early as day 9 with adult cartilage slices and only after 2 weeks with fetal explants; however, the average time to reach 80% confluence was shorter with fetal explants, though not significant (Figs 2F, 2K and 3D). Further expansion revealed spindle-shaped morphology and honeycomb growth patterns, characteristic of MSCs, with adult chondrocytes exhibiting a more fibroblastic appearance (Fig 2H). Comparison of the average cell diameter (μm) of the six groups (n = 3) showed no significant difference between them (Fig 3C). An analysis of the cell cycle distribution between the groups revealed that adult FAA-CPCs and the three fetal groups displayed a superior proliferative capacity as seen by their lower population doubling time (Fig 3A) and higher levels of the S phase (Fig 3B) when compared to the adult chondrocytes and MCP groups.

## Cell surface expression using FACS

Based on the aforementioned categories, flow cytometric analysis was performed on the six groups to assess their surface marker expression. When MSC markers were evaluated, all cell groups exhibited high expression of positive MSC markers CD105, CD73, and CD90, moderate to low expression of CD106 and a low expression of the negative markers namely CD34, CD45, and CD14 (Table 1) with no statistical difference except for CD106, which was significantly lower in the fetal chondrocytes, FAA-CPCs and MCPs group with respect to their adult counterparts (P<0.01, Fig 4A). CD106 in the fetal FAA-CPCs were significantly lower in the fetal group than in the fetal MCP group (P = 0.01). Concerning the integrin receptor marker (CD49e/CD29) expression, all the groups showed high levels except for CD49b, which was statistically higher in the adult FAA-CPCs to all groups, with comparable expression with fetal FAA-CPCs (P<0.05, Fig 4B). CD166 and Podoplanin expression were comparable among all groups; CD 146, a predictive chondrogenic marker, was significantly higher among the fetal chondrocytes, FAA-CPCs, and MCP group than in all the adult groups (P<0.001, Fig 4C). When the expression of CD146 between the adult groups was compared, chondrocytes showed significantly lower levels than the FAA-CPCs (P = 0.005) and MCPs (P = 0.038) groups. Regarding immunogenic markers, all groups displayed similar and positive expressions of class I HLA, with a minimal expression of MHC II and CD86; however, adult MCPs displayed a notably higher level of CD80 when compared to the other five groups (P< 0.01, Fig 4D).

## qRT-PCR

Regarding chondrogenesis markers, *SOX-9*, *ACAN*, and *COL2A1* showed high mRNA expression in all the groups, with no significant differences, except for *COL2A1* (Fig 5). Fetal FAA-CPCs displayed a significantly higher expression of the gene *SOX-9* when compared to the other five groups (P<0.001). However, when *ACAN* levels were assessed, adult chondrocytes and adult MCPs displayed stronger expression than all three fetal groups (P< 0.05). An analysis of hypertrophy markers revealed no differences between the six groups. A high expression of *COL1A1* and a moderate expression of *RUNX2* were observed in all the groups (Fig 5). An intergroup comparison revealed that the Fetal FAA-CPCs had remarkably higher levels of *SOX-9* (key transcription factor for the genesis of chondrocytes), and adult chondrocytes and MCPs had the highest levels of ACAN in comparison to the fetal groups. Though not

**PHASE CONTRAST**

**Fig 2. Representative images of Fetal and Adult Chondrocytes, FAA-CPCs and MCPs during monolayer expansion culture.** Microscopic analysis observed using phase-contrast microscopy showed that fetal chondrocytes (A) displayed a clonal-shaped cobblestone growth pattern similar to the FAA-CPCs (C, I), with further passage demonstrating a spindle-shape, unlike the fibroblast-like pattern observed with adult chondrocytes(H). Migration of chondroprogenitors from the cartilage explants was seen by the 10th day reaching a confluence of 80% by the third week from both fetal (F) and adult cartilage (L) explants. Magnification 10x.

significantly different, both the expression of mature type II collagen and the functional *COL2A1/COL1A1* ratio was more pronounced in the fetal and adult chondrocyte groups as compared to the adult FAA-CPCs and MCP.

## Staining following multilineage differentiation and total GAG/DNA content

Trilineage differentiation was demonstrated by all study groups. As determined through Oil Red O staining (lipid droplet accumulation) and Alizarin red S staining (calcified matrix deposition), all groups demonstrated a positive uptake of the stain (Fig 6). After chondrogenic differentiation, routine histological analyses of pellets revealed a greater uptake of Collagen type II in all fetal groups (Fig 7A1–7A3), Safranin O (Fig 7B1–7B3), and Toluidine blue (Fig 7C1–7C3) when compared to their adult counterparts. However, all groups displayed comparable Alcian Blue and Picrosirius red uptake. Measurement of the DNA and GAG contents following chondrogenic differentiation demonstrated that fetal cartilage-derived cells exhibited significantly higher levels of GAG/DNA as compared to all the adult groups (P<0.05) (Fig 8). As predicted by the gene expression analysis for chondrogenesis, higher extracellular matrix (ECM) levels were observed in the fetal groups with both immunohistochemical/routine staining and GAG/DNA content.

## Discussion and conclusion

The definitive aim of any cartilage repair strategy is to produce functional and mechanical tissue equivalents of the hyaline articular cartilage. While many reports have demonstrated the clinical efficacy of cell-based therapy, the resulting repair tissue containing hyaline, and fibro-cartilage tends not to provide long-term sustenance [2,30]. Recent studies show that cartilage resident progenitors display superior chondrogenic potential and lower hypertrophic tendencies, which have led to investigations into their possible use in tissue engineering. The standard isolation methods include clonal selection, using fibronectin adhesion, and sorting, based on the migratory assay [6]. However, with either technique, the availability to harvest cells from a sizeable sample of adult cartilage which inherently displays limited intrinsic repair, poses a significant limitation. Ideally, a single procedure, using highly proliferative allogeneic stem cells

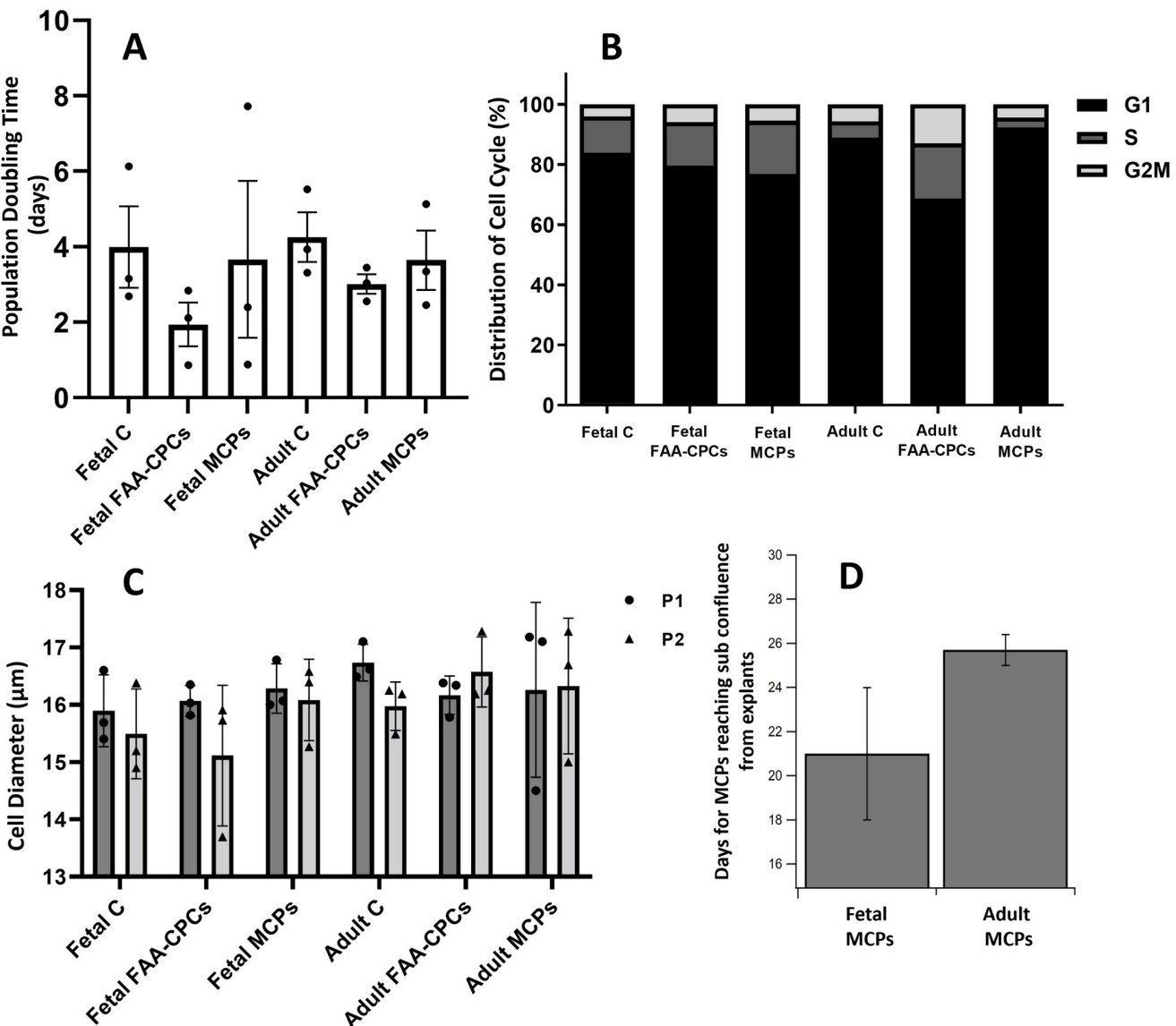

**Fig 3. Population doubling time, cell diameter and cell cycle analysis.** A) Analysis of the Population Doubling Time at passage 1 between the six groups displaying similar values with no significant difference. B) Analysis of the different phases in cell cycle distribution between the groups showed that adult FAA-CPCs that received exogenous growth factors and all three fetal groups and displayed greater proliferative capacity and higher levels of the S phase. C) Comparison of the cell diameter of the trypsinized cells at sub confluence, at passage 1 and 2 showed comparable dimensions. D) The average time in days for migration of MCPs from their explants to reach a confluence of 80% was shorter for Fetal MCPs, though their extracellular matrix displayed a dense matrix network when compared to adult cartilage.

with chondrogenic capacity and minimal osteogenic tendency, is desirable. Towards this, the attempted use of fetal cartilage-derived chondrocytes reports superior cartilage repair with positive immune modulation and is additionally being used in Phase I clinical trials for the treatment of rheumatic arthritis [27].

This study aimed to compare chondrocytes and chondroprogenitors derived using two separate but standard methods of isolation and expansion derived from fetal and adult cartilage tissue. The comparative potential of chondroprogenitors between fetal and adult sources, and in comparison to fetal chondrocytes has not been evaluated. Thus, this study aimed to compare their prospects for cartilage repair in terms of their proliferative potential, surface marker

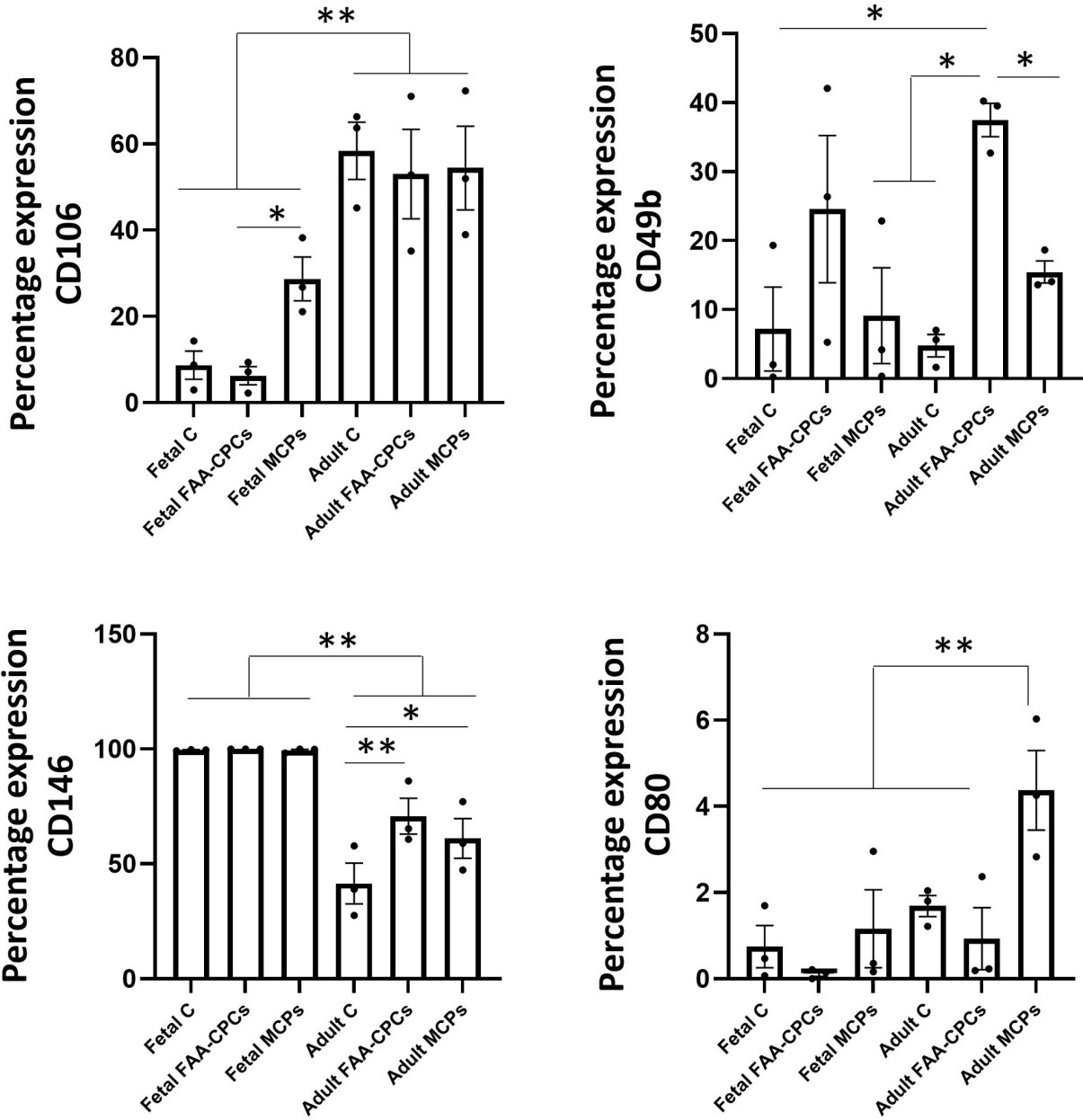

**Fig 4. Flow cytometric analysis for CD106, CD49b, CD146 and CD80.** To characterize the surface marker expression profile, Fetal and Adult Chondrocytes, FAA-CPCs and MCPs were subjected to fluorescence-activated cell sorting to check for positive and negative MSC markers, Integrin markers, markers of enhanced chondroprogenitors and immunogenic markers. All groups revealed comparable expression of CD markers (Table 1) except for CD106 and CD9b(A and B), and CD146 and CD80 (C and D). All fetal groups showed significantly lower levels of CD106 and higher levels of CD146 when compared to the adult groups. CD 49b, was noted to be considerably higher in fetal FAA-CPs. Data presented are expressed as Mean ± standard error mean. All experiments were run with three biological samples (n = 3).

expression immunophenotyping, capacity multilineage differentiation, particularly comparing their commitment for chondrogenesis using GAG/DNA analysis, routine and IHC staining for GAG and collagen proteins and mRNA expression analysis of primary markers of chondrogenesis and hypertrophy.

The process of chondrogenesis is positively correlated with cellular morphological changes. In this study, we show that the isolation of cartilage resident cells was possible using similar

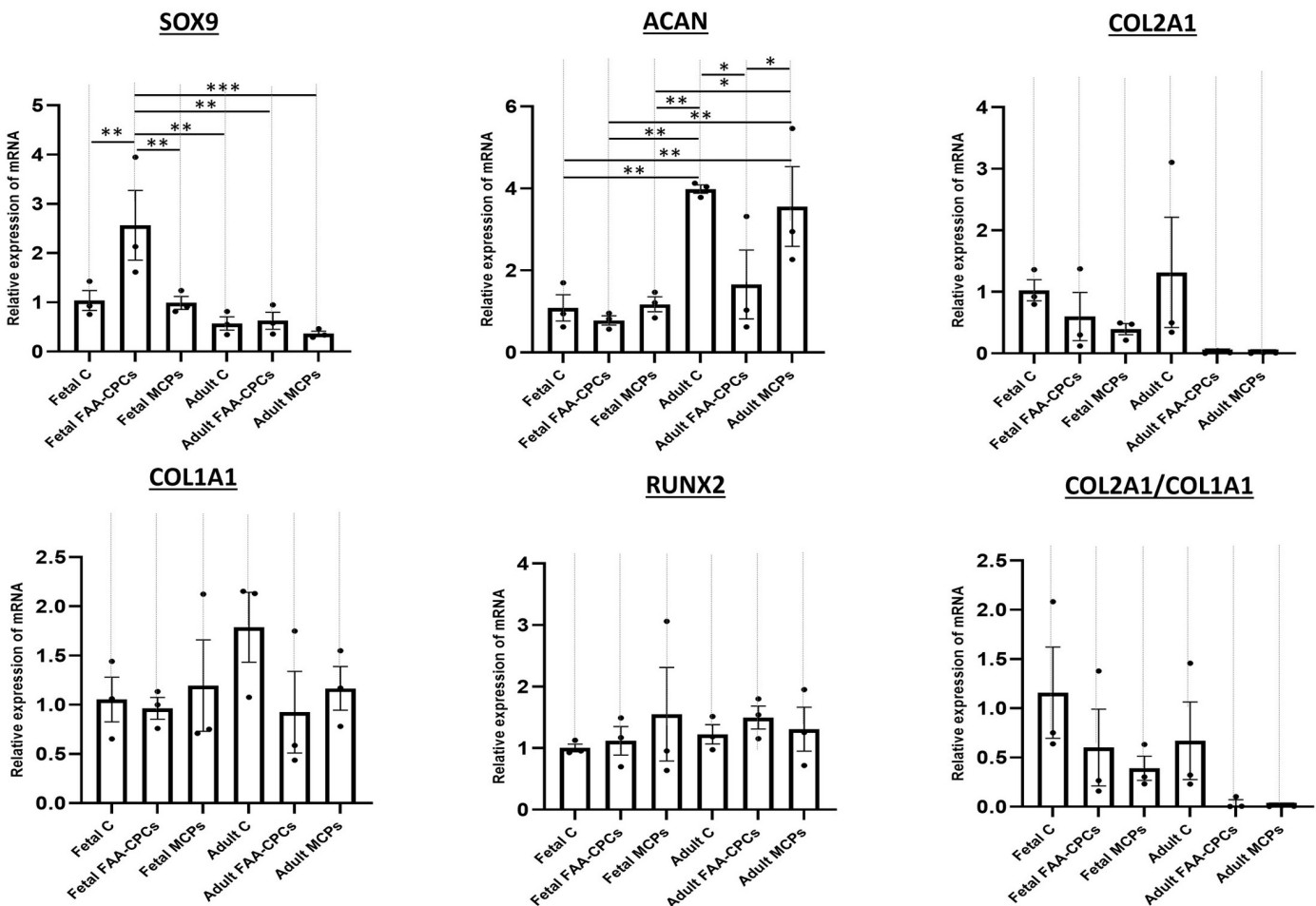

**Fig 5. Relative mRNA expression encoding chondrogenic markers (SOX-9, ACAN, COL2A1), fibrocartilage marker (COL1A1), and hypertrophic markers (RUNX2) were determined by qRT-PCR.** All results were normalized to the housekeeping gene, GAPDH(ΔCt), and ΔΔCt was obtained in comparison to the Fetal Chondrocyte group. 2^-ΔΔCt values are expressed as mean ± standard error mean (*P<0.05, ** P<0.01, ***P< 0.001). All experiments were run with three biological samples (n = 3) in two technical replicates. Overall, the Fetal FAA-CPCs displayed higher levels of SOX-9 when compared to the other groups. All groups displayed constitutive expression of ACAN with the adult chondrocytes and MCPs expressed significantly higher levels. s groups.

isolation techniques with both fetal and adult tissue. Although the fetal tissue was more accessible, it displayed a denser matrix containing a more significant cell-to-matrix ratio when compared to the adult tissue, thus requiring a longer time for the start of migration of MCPs. Concerning the morphology, the fetal chondrocytes displayed a clonal-shaped cobblestone growth pattern similar to the FAA-CPCs, with further passage demonstrating a spindle-shaped, unlike the fibroblast-like pattern observed with adult chondrocytes. The changes in the cellular level originate from events at the molecular level. As previously reported, our study indicated that adult FAA-CPCs that received exogenous growth factors and all three fetal groups displayed greater proliferative capacity and higher levels of the S phase, a vital requirement for accurate genome duplication critical for successful cell division [24,31]. Our results also showed that the time for fetal MCPs to migrate from their dense matrix was shorter than adult MCPs, with the fetal MCPs showed cell cycle patterns similar to FAA-CPCs, even though the culture conditions did not include additional exogenous growth factors. Comparing the

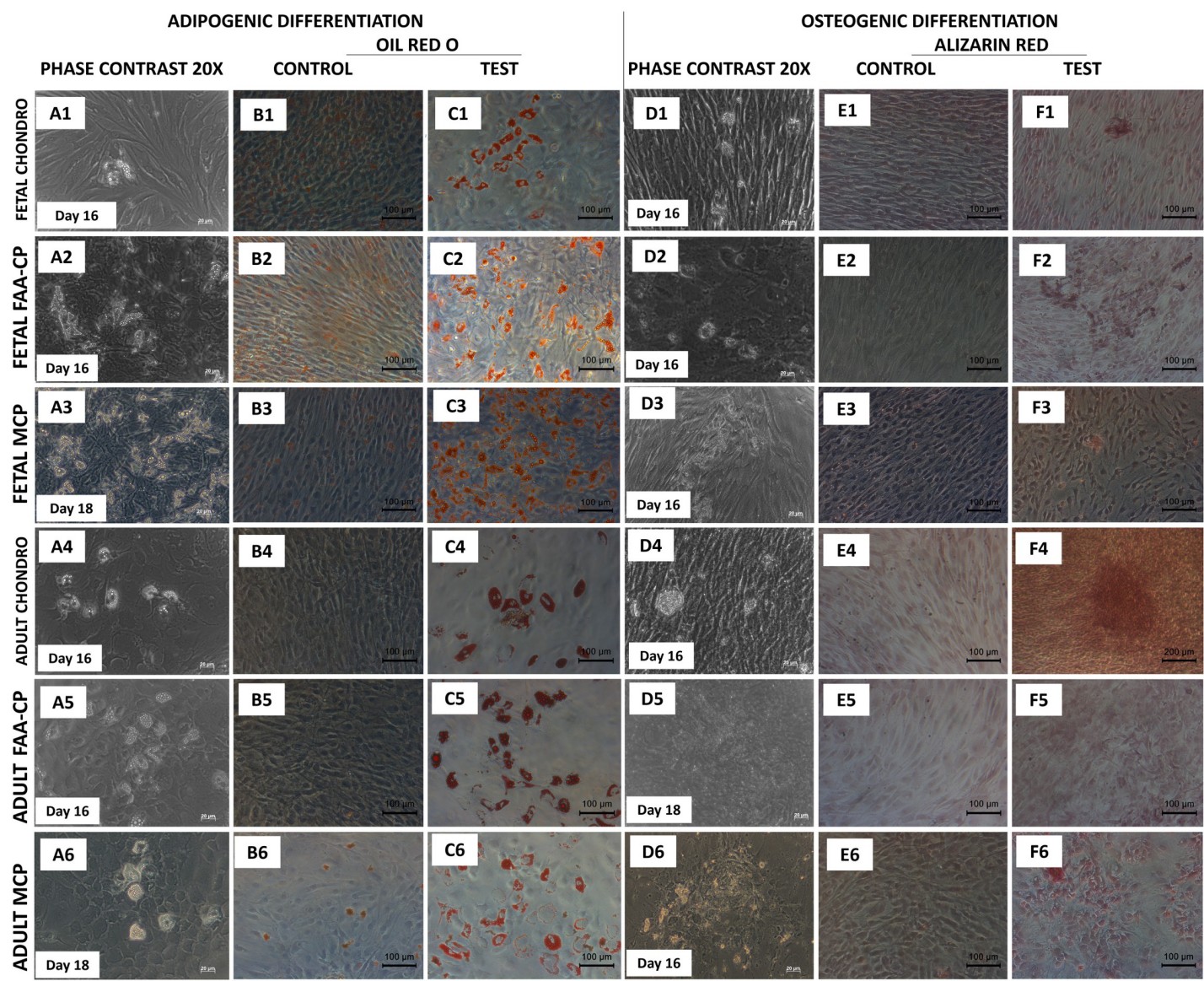

**Fig 6. Representative phase contrast images and confirmatory staining following adipogenic and osteogenic differentiation.** Unstained cells at day 18 during adipogenic (A1-A6) and osteogenic (D1-D6) differentiation and their confirmatory staining using Oil Red O (C1-C6) and Alizarin red (F1-F6). Magnification: 20X. All groups showed uptake of the Oil Red O stain indicative of lipid droplet accumulation and uptake of Alizarin red confirming calcium deposition, with the adult chondrocyte group displaying greater staining. Controls in all groups did not show any uptake of stain (B1-B6, E1-E6).

population doubling time between the groups, the fetal CPCs required the least time to reach sub confluence.

The immunophenotyping expression pattern showed that all groups complied with the ISCT minimal criteria, with significant differences observed for CD106, CD146, CD49b and CD80. CD106 is a marker for mesenchymal stem cells, which has also been reported to be a predictive marker for osteogenesis, while CD146 is a putative marker for enhanced chondrogenesis [32–34]. It was noted that all fetal groups showed significantly lower levels of CD106 and higher levels of CD146 when compared to the adult groups. The gene expression profile analysis showed that all the fetal groups showed higher expression of *SOX-9*, a primary transcription factor for chondrogenesis, with the fetal FAA-CPCs displaying the most elevated

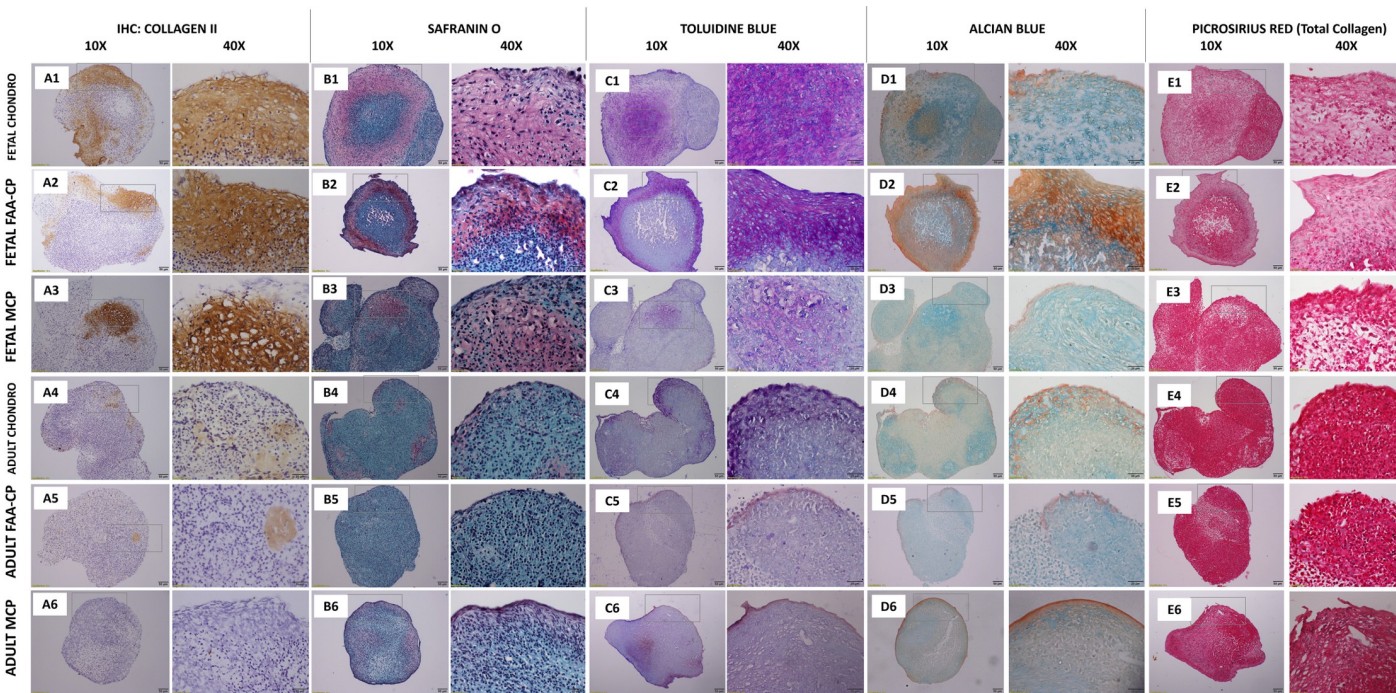

**Fig 7. Representative images of chondrogenic differentiated pellets of the six groups: Fetal and Adult Chondrocytes, FAA-CPCs and MCPs.** The formed pellets (1x106) were grown in StemPro chondrogenic medium for a period of 21 days. Greater uptake of Collagen type II (A1-A3), Safranin O (B1-B3), Toluidine blue (C1-C3) was observed with the Fetal cartilage derived cells when compared to their adult counterparts. All groups displayed comparable Alcian Blue and Picrosirius red uptake: Total collagen content Magnification: Scale bar (50μm or 20μm).

levels significantly. This finding corroborated to a report by Choi et al. who reported a higher expression of the primary chondrogenic factor, *SOX-9*, by fetal chondrocytes as compared to BM-MSCs [24]. When the expression of *ACAN*, a major articular ECM component was compared, all groups showed a constitutive expression; however, adult chondrocytes and MCPs expressed significantly higher levels. It is essential to emphasize that the RT-PCR comparison is based on the fact that the analysis was performed on undifferentiated cells grown in their recommended standard media. The gold standard technique to infer the potential in-vivo behaviour of the cells includes driving a three-dimensional culture towards chondrogenic differentiation using a single formulation rich in growth factors [35]. Analysis of the GAG/DNA ratio showed that fetal chondrocytes, fetal FAA-CPCs and fetal MCPs outperformed the other adult cell groups significantly. Across all six groups, the retention of multipotency was confirmed by the results of trilineage differentiation. Compared to the adult groups, all fetal groups exhibited a significant and higher deposition of the collagen type II. Moreover, there was a more substantial uptake for glycosaminoglycans as seen with Alcian blue, Safranin O and Toluidine blue.

It was noted that the uptake of alizarin red was greater with the adult chondrocyte groups, which corroborated with the higher *COL1A1* gene expression levels when compared to the other groups. CD 49b, an established integrin differentiating marker for MCPs, was noted to be considerably higher in fetal FAA-CPCs. CD80, a co-stimulatory MHC molecule marker was significantly higher with the adult MCPs group, implying the need for further evaluation.

Fetal cartilage-derived resident chondrocytes and progenitors are easier to isolate and expand than adult tissue due to their abundance in cartilage-to-bone ratio. Additionally, they possess stem cell traits such as self-renewal, multipotent differentiation, and low immunogenic

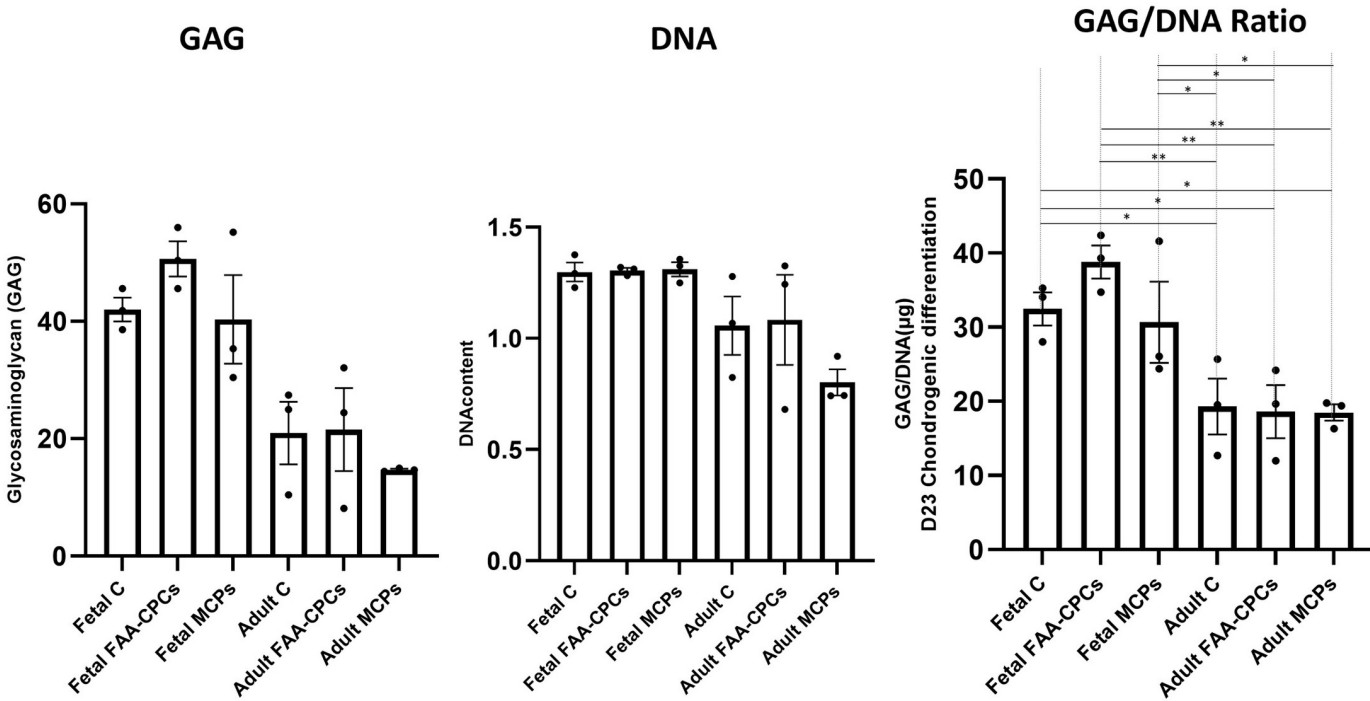

**Fig 8. Measurement of the total GAG, DNA and GAG/DNA content of differentiated chondrogenic pellets.** Fetal Chondrocytes, FAA-CPCs and MCPs demonstrated significantly higher levels of GAG/DNA when compared to their adult counterparts. All values are expressed as Mean ± standard error mean (*P<0.05, ** P<0.01, ***P< 0.001), from three biological samples (n = 3).

marker profile, making them a safe allogeneic cell source. Thirdly, the ability of fetal cells to form cartilage tissue outperformed the adult cells, with the former displaying more GAG and collagen type II production as compared to the adult sources. A comparison between the fetal groups revealed fetal FAA-CPCs, demonstrating more remarkable proliferation ability with higher *SOX-9* expression.

This study is the first in-vitro report to establish and study the expansion of fetal chondrocytes and chondroprogenitors as compared to their adult counterparts. As demonstrated in this study, fetal tissue displayed a higher propensity for chondrogenesis in a fundamentally superior manner than adult tissue. Though the cells isolated from adult cartilage were from intact seemingly normal tissue, these cells may have been subject to inflammatory changes and this may affect its properties. Our findings support previously published studies reporting the superiority of fetal chondrocytes as translational cellular candidates for cartilage repair, now with fetal chondroprogenitors as a potential cell type [24,26]. Although, our results show that fetal resident cartilage cells possess a higher chondrogenic potential, a comparative surfaceome and proteomic analysis, revalidation using in- vivo study models, and studying cells obtained from non-diseased joints instead of osteoarthritic joints would further the understanding of their potential therapeutic application in cartilage repair.

## Supporting information

**S1 Table. List of antibodies used for characterization of chondroprogenitors by flow cytometric analysis.** MSC: Mesenchymal stem cell, CD: Cluster of differentiation, FITC: Fluorescein isothiocyanate, PE: Phycoerythrin, APC: Allophycocyanin, BB515: Horizon brilliant blue

515, BV421: Brilliant violet 421 and V500: Violet 500.
(DOCX)

**S2 Table. Sequence of the primers used for RT-PCR.** SOX9: Sex determining region Y-box 9, *ACAN*: Aggrecan, *COL2A1*: Collagen type 2 alpha 1 chain, *COL1A1*: Collagen type 1 alpha 1 chain, *COL10A1*: Collagen type 10 alpha 1 chain, *RUNX2*: Runt related transcription factor-2 and *GAPDH*: Glyceraldehyde 3-phosphate dehydrogenase.
(DOCX)

**S1 Dataset.**
(XLSX)

# Acknowledgments

We acknowledge Dr. Pippa Deodhar, Scientist, Research Promotion and Development, Principal's Office, CMC Vellore, for help with editing the manuscript, Mr Abdul Muthallib, Ms Esther Rani and Mr Ashok Kumar for technical support, and the Centre for Stem Cell Research (A unit of inStem Bengaluru), Christian Medical College, Vellore for infrastructural support. The research work presented in this manuscript is a part of the PhD thesis of The Tamil Nadu Dr. M.G.R. Medical University.

# Author Contributions

**Conceptualization:** Elizabeth Vinod, Boopalan Ramasamy, Solomon Sathishkumar.

**Data curation:** Elizabeth Vinod, Ganesh Parasuraman, Jeya Lisha J., Soosai Manickam Amirtham, Abel Livingston, Sandya Rani, Deepak Vinod Francis, Alfred Job Daniel.

**Formal analysis:** Elizabeth Vinod, Ganesh Parasuraman, Jeya Lisha J., Soosai Manickam Amirtham, Abel Livingston, Jithu James Varghese, Sandya Rani, Deepak Vinod Francis, Grace Rebekah, Alfred Job Daniel, Boopalan Ramasamy, Solomon Sathishkumar.

**Funding acquisition:** Elizabeth Vinod, Soosai Manickam Amirtham.

**Investigation:** Elizabeth Vinod, Jeya Lisha J., Soosai Manickam Amirtham, Abel Livingston, Sandya Rani, Deepak Vinod Francis, Grace Rebekah.

**Methodology:** Elizabeth Vinod, Ganesh Parasuraman, Jeya Lisha J., Soosai Manickam Amirtham, Abel Livingston, Jithu James Varghese, Sandya Rani, Deepak Vinod Francis, Grace Rebekah, Alfred Job Daniel, Solomon Sathishkumar.

**Project administration:** Elizabeth Vinod.

**Resources:** Elizabeth Vinod.

**Software:** Grace Rebekah.

**Supervision:** Elizabeth Vinod, Boopalan Ramasamy.

**Validation:** Elizabeth Vinod, Ganesh Parasuraman, Abel Livingston, Jithu James Varghese, Deepak Vinod Francis, Grace Rebekah, Boopalan Ramasamy, Solomon Sathishkumar.

**Writing – original draft:** Elizabeth Vinod, Ganesh Parasuraman, Jeya Lisha J., Soosai Manickam Amirtham, Abel Livingston, Jithu James Varghese, Sandya Rani, Deepak Vinod Francis, Grace Rebekah, Alfred Job Daniel, Boopalan Ramasamy, Solomon Sathishkumar.

**Writing – review & editing:** Elizabeth Vinod, Boopalan Ramasamy, Solomon Sathishkumar.

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
