## [Decision Letter · Decision Letter 0]

7 Feb 2023

PONE-D-22-32889Human fetal cartilage-derived chondrocytes and chondroprogenitors display a greater commitment to chondrogenesis than adult cartilage resident cellsPLOS ONE

Dear Dr. Vinod,

Thank you for submitting your manuscript to PLOS ONE. After careful consideration, we feel that it has merit but does not fully meet PLOS ONE’s publication criteria as it currently stands. Therefore, we invite you to submit a revised version of the manuscript that comprehensively addresses all the points raised by the two reviewers.

We look forward to receiving your revised manuscript.

Kind regards,

Andre van Wijnen

Academic Editor

PLOS ONE

Journal Requirements:

When submitting your revision, we need you to address these additional requirements. 1. Please ensure that your manuscript meets PLOS ONE's style requirements, including those for file naming. The PLOS ONE style templates can be found at https://journals.plos.org/plosone/s/file?id=wjVg/PLOSOne_formatting_sample_main_body.pdf and https://journals.plos.org/plosone/s/file?id=ba62/PLOSOne_formatting_sample_title_authors_affiliations.pdf 2. Thank you for stating in your Funding Statement: "This project was supported by the 1.
Department of Biotechnology (BT/PR32777/MED/31/415/2019), Govt. of India Recipient: E.V2.
Fluid Research Grant (IRB Min No: 14498 dated 23.02.2022), Christian Medical College, Vellore.  Recipient: S.M.A" Please provide an amended statement that declares *all* the funding or sources of support (whether external or internal to your organization) received during this study, as detailed online in our guide for authors at http://journals.plos.org/plosone/s/submit-now.  Please also include the statement “There was no additional external funding received for this study.” in your updated Funding Statement. Please include your amended Funding Statement within your cover letter. We will change the online submission form on your behalf. 3. Thank you for stating the following financial disclosure: "This project was supported by the 1.
Department of Biotechnology (BT/PR32777/MED/31/415/2019), Govt. of India Recipient: E.V2.
Fluid Research Grant (IRB Min No: 14498 dated 23.02.2022), Christian Medical College, Vellore.  Recipient: S.M.A" Please state what role the funders took in the study. If the funders had no role, please state: ""The funders had no role in study design, data collection and analysis, decision to publish, or preparation of the manuscript."" If this statement is not correct you must amend it as needed. Please include this amended Role of Funder statement in your cover letter; we will change the online submission form on your behalf. 4. In your Data Availability statement, you have not specified where the minimal data set underlying the results described in your manuscript can be found. PLOS defines a study's minimal data set as the underlying data used to reach the conclusions drawn in the manuscript and any additional data required to replicate the reported study findings in their entirety. All PLOS journals require that the minimal data set be made fully available. For more information about our data policy, please see http://journals.plos.org/plosone/s/data-availability. Upon re-submitting your revised manuscript, please upload your study’s minimal underlying data set as either Supporting Information files or to a stable, public repository and include the relevant URLs, DOIs, or accession numbers within your revised cover letter. For a list of acceptable repositories, please see http://journals.plos.org/plosone/s/data-availability#loc-recommended-repositories. Any potentially identifying patient information must be fully anonymized. Important: If there are ethical or legal restrictions to sharing your data publicly, please explain these restrictions in detail. Please see our guidelines for more information on what we consider unacceptable restrictions to publicly sharing data: http://journals.plos.org/plosone/s/data-availability#loc-unacceptable-data-access-restrictions. Note that it is not acceptable for the authors to be the sole named individuals responsible for ensuring data access. We will update your Data Availability statement to reflect the information you provide in your cover letter. 5. Please include your full ethics statement in the ‘Methods’ section of your manuscript file. In your statement, please include the full name of the IRB or ethics committee who approved or waived your study, as well as whether or not you obtained informed written or verbal consent. If consent was waived for your study, please include this information in your statement as well.  6. We note that Figure 1 in your submission contain copyrighted images. All PLOS content is published under the Creative Commons Attribution License (CC BY 4.0), which means that the manuscript, images, and Supporting Information files will be freely available online, and any third party is permitted to access, download, copy, distribute, and use these materials in any way, even commercially, with proper attribution. For more information, see our copyright guidelines: http://journals.plos.org/plosone/s/licenses-and-copyright. We require you to either (1) present written permission from the copyright holder to publish these figures specifically under the CC BY 4.0 license, or (2) remove the figures from your submission: a. You may seek permission from the original copyright holder of Figure 1 to publish the content specifically under the CC BY 4.0 license.  We recommend that you contact the original copyright holder with the Content Permission Form (http://journals.plos.org/plosone/s/file?id=7c09/content-permission-form.pdf) and the following text:“I request permission for the open-access journal PLOS ONE to publish XXX under the Creative Commons Attribution License (CCAL) CC BY 4.0 (http://creativecommons.org/licenses/by/4.0/). Please be aware that this license allows unrestricted use and distribution, even commercially, by third parties. Please reply and provide explicit written permission to publish XXX under a CC BY license and complete the attached form.” Please upload the completed Content Permission Form or other proof of granted permissions as an ""Other"" file with your submission.  In the figure caption of the copyrighted figure, please include the following text: “Reprinted from [ref] under a CC BY license, with permission from [name of publisher], original copyright [original copyright year].” b. If you are unable to obtain permission from the original copyright holder to publish these figures under the CC BY 4.0 license or if the copyright holder’s requirements are incompatible with the CC BY 4.0 license, please either i) remove the figure or ii) supply a replacement figure that complies with the CC BY 4.0 license. Please check copyright information on all replacement figures and update the figure caption with source information. If applicable, please specify in the figure caption text when a figure is similar but not identical to the original image and is therefore for illustrative purposes only. 7. Please include captions for your Supporting Information files at the end of your manuscript, and update any in-text citations to match accordingly. Please see our Supporting Information guidelines for more information: http://journals.plos.org/plosone/s/supporting-information. 

Reviewers' comments:

Reviewer's Responses to Questions

**Comments to the Author**

1. Is the manuscript technically sound, and do the data support the conclusions?

Reviewer #1: Partly

Reviewer #2: Yes

2. Has the statistical analysis been performed appropriately and rigorously? 

Reviewer #1: Yes

Reviewer #2: Yes

3. Have the authors made all data underlying the findings in their manuscript fully available?

Reviewer #1: No

Reviewer #2: Yes

4. Is the manuscript presented in an intelligible fashion and written in standard English?

Reviewer #1: Yes

Reviewer #2: Yes

5. Review Comments to the Author

Reviewer #1: This study looks at six different cell populations isolated from either fetal or adult cartilage using 3 different methods. The cells are compared for growth kinetics, marker profile, gene expression analysis, and tri-lineage differentiation. The premise for the study is that the comparative potential of chondroprogenitors between fetal and adult sources, and in

comparison to fetal chondrocytes has not been evaluated. The goal of the study is interesting, and this is a valuable study, however there were some significant concerns with some of the experimental design and interpretation of the data. Specific comments are outlined below:

Adult cartilage was isolated from total knee replacement patients with high grade OA. The authors should describe where they isolated the cartilage from, as there is often limited cartilage available from these samples and it may not represent a normal cartilage.

Figure 2 – Growth kinetics: the days that the images were taken at are not standardized so it is very difficult to compare. Similar days should be compared to really assess differences. The text (line 286) mentions days 11-12 and figure is day 10-11.

Figure 3A – Please provide a description of how population doubling time was calculated. What is the unit of time (hrs, days)? When was this population doubling measured (P0 to P1 or P1 to P2)?

Figure 3D – there is a lot of mention of subconfluence in the results but the manner in which it is determined is not clear to the reader. This makes it difficult to appreciate any differences seen.

Figure 4 – statistical differences are mentioned in text but not shown in the figure. Need to show quantitative data, perhaps as bar graphs. Some markers were mentioned in text but not shown in Figure. It made the reader wonder whether the wrong figure was uploaded. Please verify this data and section of manuscript.

Line 359-360: the authors state that the fetal groups show higher expression of Sox9 but only one population was statistically significant. The authors should be careful to not overstate the results.

Minor

Pg15, line 190: can remove “the blue fluorescent DAPI stain”. Starting the sentence at DAPI was used…. Should be sufficient

Methods, line 256-250: there is a lot of description of ethanol solutions used. Some details can be removed and replaced with processed for paraffin embedding

Line 390: typo in the word fetal

Reviewer #2: This is a useful study that adds to the body of work that describes the properties of various sub-populations of cells in articular cartilage. Ultimately the search for the ideal ‘cell’ for cartilage repair relies on this type of comparative analyses. The authors do not address any negative aspects of their study in the Discussion the foremost of which is the use of adult cells taken from individuals who had osteoarthritis – the potential deficiencies of the study need to be highlighted. Whilst the cartilage used may be intact the cells will have been subject to inflammatory insults and this may affect their properties. The results from intact human cartilage – which is difficult to obtain – may have given similar results as the fetal derivatives.

Review: Vinod et al

L81 – surely you mean ‘dedifferentiation’.

L88 – delete 2nd instance of ‘MSC’.

L90 - Fibronectin adhesion is not selective isolation of chondroprogenitors - it is 'enrichment for colony forming cells'. Selective implies a process where a specific subset of cells is being isolated, this isn't the case, it is an enrichment process where the probability of isolating 'progenitor cell' is increased.

L110 – “collagen type II’.

L141 – in the M&M it’s clear you are using polyclonal cells, so when you use the word ‘clonal’ you are confusing readers.

L215 - M&M qPCR which sample was used as the calibrator for this study, please state as this is standard practice when using relative quantification.

6. PLOS authors have the option to publish the peer review history of their article (what does this mean?). If published, this will include your full peer review and any attached files.

Reviewer #1: No

Reviewer #2: No

---

## [Author Response · Author response to Decision Letter 0]

21 Mar 2023

Authors would like to sincerely thank the reviewers for taking the time to go through the manuscript and providing a detailed review. We appreciate the opportunity to provide clarifications and modify our manuscript accordingly.

Reviewer #1: This study looks at six different cell populations isolated from either fetal or adult cartilage using 3 different methods. The cells are compared for growth kinetics, marker profile, gene expression analysis, and tri-lineage differentiation. The premise for the study is that the comparative potential of chondroprogenitors between fetal and adult sources, and in comparison to fetal chondrocytes has not been evaluated. The goal of the study is interesting, and this is a valuable study, however there were some significant concerns with some of the experimental design and interpretation of the data. 

The authors thank the reviewer for the suggestions and have made the required modifications to the manuscript. 

Specific comments are outlined below:

Adult cartilage was isolated from total knee replacement patients with high grade OA. The authors should describe where they isolated the cartilage from, as there is often limited cartilage available from these samples and it may not represent a normal cartilage.

The manuscript has been modified to include the required details. The articular cartilage was harvested from non-weight bearing areas, with preserved full depth cartilage. (Line No: 133-134) 

Figure 2 – Growth kinetics: the days that the images were taken at are not standardized so it is very difficult to compare. Similar days should be compared to really assess differences. The text (line 286) mentions days 11-12 and figure is day 10-11.

Figure 2 is only the representative images of Fetal and Adult Chondrocytes, FAA-CPCs and MCPs during monolayer expansion culture. In order to show similar confluence but also the variability based on the tissue, the particular days were chosen. The migration of the progenitors from the fetal and adult explants started at variable timepoints.

Concerning the days mentioned in the text, the authors regret this error and the writeup has been duly modified. (Line no: 293-294)

Figure 3A – Please provide a description of how population doubling time was calculated. What is the unit of time (hrs, days)? When was this population doubling measured (P0 to P1 or P1 to P2)?

The authors regret the oversight, the writeup and the figure axis have been modified including details on the unit of time (days), the formulae used and the passage number at which it was measured (P1 to P2). (Line no: 191-197) 

Figure 3D – there is a lot of mention of subconfluence in the results but the manner in which it is determined is not clear to the reader. This makes it difficult to appreciate any differences seen.

The figure legend has been modified mentioning the percentage used to determine the sub confluence. The authors have reworded the term sub confluence to 80% confluence or sub confluence (80% confluence). (Line no: 142-143, 296, 309-310 and 318). 

Figure 4 – statistical differences are mentioned in text but not shown in the figure. Need to show quantitative data, perhaps as bar graphs. Some markers were mentioned in text but not shown in Figure. It made the reader wonder whether the wrong figure was uploaded. Please verify this data and section of manuscript.

As suggested this figure (Figure 4) has been presented as a bar graph including the significant differences between the groups. Fluorescence-activated cell sorting data for positive and negative MSC markers, potential markers of enhanced chondrogenesis, and immunogenic markers of the six cell groups has been presented in a tabular format (Table 1) as percentage mean ± Standard Deviation (n=3). However, the values showing significant difference between the groups were only used for the histogram plot. 

Line 359-360: the authors state that the fetal groups show higher expression of Sox9 but only one population was statistically significant. The authors should be careful to not overstate the results.

The authors thank the reviewer for this observation. The manuscript has been modified stating only the significant difference. (Line no: 367-368). 

Minor

Pg15, line 190: can remove “the blue fluorescent DAPI stain”. Starting the sentence at DAPI was used…. Should be sufficient

Suggested modification has been included. (Line no: 198)

Methods, line 256-250: there is a lot of description of ethanol solutions used. Some details can be removed and replaced with processed for paraffin embedding

The writeup has been modified as suggested. (Line no: 256-257)

Line 390: typo in the word fetal

The authors regret this oversight, the typo has been corrected. (Line No: 398)

Reviewer # 2: 

This is a useful study that adds to the body of work that describes the properties of various sub-populations of cells in articular cartilage. Ultimately the search for the ideal ‘cell’ for cartilage repair relies on this type of comparative analyses. The authors do not address any negative aspects of their study in the Discussion the foremost of which is the use of adult cells taken from individuals who had osteoarthritis – the potential deficiencies of the study need to be highlighted. Whilst the cartilage used may be intact the cells will have been subject to inflammatory insults and this may affect their properties. The results from intact human cartilage – which is difficult to obtain – may have given similar results as the fetal derivatives.

The authors thank the reviewer for this suggestion. The writeup has been modified to include the limitations of the study. (Line no: 479-481, 483-487)

L81 – surely you mean ‘dedifferentiation’.

The authors regret the oversight and the following corrections have been duly made

The change has been done. Line no: 77

L88 – delete 2nd instance of ‘MSC’.

The change has been done. Line no: 84

L90 - Fibronectin adhesion is not selective isolation of chondroprogenitors - it is 'enrichment for colony forming cells'. Selective implies a process where a specific subset of cells is being isolated, this isn't the case, it is an enrichment process where the probability of isolating 'progenitor cell' is increased.

The change has been done. Line no: 86

L110 – “collagen type II’.

The change has been done. Line no: 106

L141 – in the M&M it’s clear you are using polyclonal cells, so when you use the word ‘clonal’ you are confusing readers.

The change has been done. Line no: 140

L215 - M&M qPCR which sample was used as the calibrator for this study, please state as this is standard practice when using relative quantification.

The change has been done. Line 233-234

Journal Requirements:

The authors have modified the manuscript ensuring that the PLOS ONE's style requirements are met. 

Modifications have been applied for the following: 

- Level 1, 2, 3 headings

- Figure and table captions

- Title page

"This project was supported by the 

1. Department of Biotechnology (BT/PR32777/MED/31/415/2019), Govt. of India 

Recipient: E.V

2. Fluid Research Grant (IRB Min No: 14498 dated 23.02.2022), Christian Medical College, Vellore. 

Recipient: S.M.A"

The required details have been included in the financial disclosure document and the cover letter. 

"This project was supported by the 

1. Department of Biotechnology (BT/PR32777/MED/31/415/2019), Govt. of India 

Recipient: E.V

2. Fluid Research Grant (IRB Min No: 14498 dated 23.02.2022), Christian Medical College, Vellore. 

Recipient: S.M.A"

The required details have been included in the financial disclosure document and the cover letter.

The authors confirm that all data underlying the findings are fully available without restriction. All relevant data are within the paper. Additionally, the minimal data set (excel sheet) has been now uploaded as a supporting information file. (S1 Data set).

The full ethics statement has been included in the methods section. (Line no: 127-131, 135-137)

6. We note that Figure 1 in your submission contain copyrighted images. All PLOS content is published under the Creative Commons Attribution License (CC BY 4.0), which means that the manuscript, images, and Supporting Information files will be freely available online, and any third party is permitted to access, download, copy, distribute, and use these materials in any way, even commercially, with proper attribution. For more information, see our copyright guidelines: http://journals.plos.org/plosone/s/licenses-and-copyright.

All the images were created by the authors for the purpose of this study and does not include any copyright material. However, we have modified two images and additionally mention in the figure caption the following statement: 

The included figure is similar but not identical to the original image and is for illustrative purposes only. (Line no: 157-158).

The captions for Supporting Information files have been included at the end of the manuscript, ensuring matched in-text citations.

---

## [Decision Letter · Decision Letter 1]

13 Apr 2023

PONE-D-22-32889R1Human fetal cartilage-derived chondrocytes and chondroprogenitors display a greater commitment to chondrogenesis than adult cartilage resident cellsPLOS ONE

Dear Dr. Vinod,

Thank you for submitting your manuscript to PLOS ONE. After careful consideration, we feel that it has merit and meets the publication criteria of PLOS ONE. Therefore, we invite you to submit a revised version of the manuscript that addresses the final recommendations for text revisions. These points represent minor revisions and hence please consider your paper provisionally accepted pending these final modifications. 

We look forward to receiving your revised manuscript.

Kind regards,

Andre van Wijnen, PhD

Academic Editor

PLOS ONE

Journal Requirements:

Additional Editor Comments:

This paper is essentially provisionally accepted pending very minor final revisions of the paper.

Reviewers' comments:

Reviewer's Responses to Questions

**Comments to the Author**

1. If the authors have adequately addressed your comments raised in a previous round of review and you feel that this manuscript is now acceptable for publication, you may indicate that here to bypass the “Comments to the Author” section, enter your conflict of interest statement in the “Confidential to Editor” section, and submit your "Accept" recommendation.

Reviewer #1: (No Response)

2. Is the manuscript technically sound, and do the data support the conclusions?

Reviewer #1: Partly

3. Has the statistical analysis been performed appropriately and rigorously? 

Reviewer #1: Yes

4. Have the authors made all data underlying the findings in their manuscript fully available?

Reviewer #1: Yes

5. Is the manuscript presented in an intelligible fashion and written in standard English?

Reviewer #1: Yes

6. Review Comments to the Author

Reviewer #1: The reviewers have addressed all comments and have some minor edits that would benefit the manuscript.

1. Figure 7: the images in the SafraninO column look like Alcian Blue and the images in the Alcian Blue column look like Safranin O. Please check and correct if necessary

2. line 110: remove the work "remarkable" . It is not needed.

3. line 415: need a reference about the Phase I clinical trial mentioned.

4. line 483: would benefit from some references

7. PLOS authors have the option to publish the peer review history of their article (what does this mean?). If published, this will include your full peer review and any attached files.

Reviewer #1: No

---

## [Author Response · Author response to Decision Letter 1]

14 Apr 2023

Authors would like to sincerely thank the reviewers for taking the time to go through the manuscript and providing a detailed review. We appreciate the opportunity to provide clarifications and modify our manuscript accordingly.

Additional Editor Comments

1. This paper is essentially provisionally accepted pending very minor final revisions of the paper.

The authors thank the editor for this comment and have made the required modifications to the manuscript. 

Reviewer #1:

The reviewers have addressed all comments and have some minor edits that would benefit the manuscript.

The authors thank the reviewer for the suggestions and have made the required modifications to the manuscript. 

1.Figure 7: the images in the SafraninO column look like Alcian Blue and the images in the Alcian Blue column look like Safranin O. Please check and correct if necessary

The authors have rechecked the figures and ensured that the grid is correct. Safranin O staining was performed using 1% Safranin O solution with fast green. The pellets displayed a pink to orange uptake at the areas showing the presence of proteoglycans. 

In the case of Alcian blue staining, it was performed at a pH of 2.5 with a neutral red counterstain, staining areas containing sulfated and carboxylated glycosaminoglycans (blue to turquoise coloration). (Line no: 259-262).

2. line 110: remove the work "remarkable" . It is not needed.

As suggested the word remarkable has been removed. (Line No: 110)

3. line 415: need a reference about the Phase I clinical trial mentioned.

Thank you for the suggestion, the reference has been included. (Line no: 416) 

4. line 483: would benefit from some references

Relevant references have been included (Line No: 484)

Journal Requirements:

Please review your reference list to ensure that it is complete and correct. If you have cited papers that have been retracted, please include the rationale for doing so in the manuscript text,or remove these references and replace them with relevant current references. Any changes to the reference list should be mentioned in the rebuttal letter that accompanies your revised manuscript. If you need to cite a retracted article, indicate the article’s retracted status in the References list and also include a citation and full reference for the retraction notice.

The authors have crosschecked the cited papers, ensuring it does not include any retracted papers. 

We found that two references from the same paper were duplicated. We regret this oversight, and we have rectified the issue.

---

## [Editor Report · Decision Letter 2]

17 Apr 2023

Human fetal cartilage-derived chondrocytes and chondroprogenitors display a greater commitment to chondrogenesis than adult cartilage resident cells

PONE-D-22-32889R2

Dear Dr. Vinod,

We’re pleased to inform you that your manuscript has been judged scientifically suitable for publication and will be formally accepted for publication once it meets all outstanding technical requirements.

Kind regards,

Andre van Wijnen, PhD

Academic Editor

PLOS ONE

Additional Editor Comments (optional):

Editorial comments: the authors have adequately addressed the final minor comments.
---

## [Editor Report · Acceptance letter]

19 Apr 2023

PONE-D-22-32889R2 

Human fetal cartilage-derived chondrocytes and chondroprogenitors display a greater commitment to chondrogenesis than adult cartilage resident cells 

Dear Dr. Vinod:

I'm pleased to inform you that your manuscript has been deemed suitable for publication in PLOS ONE. Congratulations! Your manuscript is now with our production department. 

Kind regards, 

on behalf of

Dr. Andre van Wijnen 

Academic Editor

PLOS ONE